# A Multidisciplinary Approach for the Characterization of Artificial Cavities of Historical and Cultural Interest: The Case Study of the Cloister of Sant'Agostino—Caserta, Italy

Emilia Damiano [1,*], Maria Assunta Fabozzi [1], Paolo Maria Guarino [2], Erika Molitierno [1], Lucio Olivares [1], Roberto Pratelli [3], Marco Vigliotti [1] and Daniela Ruberti [1]

[1] Department of Engineering, University of Campania "Luigi Vanvitelli", 81031 Aversa, Italy; mariaassunta.fabozzi@unicampania.it (M.A.F.); erika.molitierno@unicampania.it (E.M.); lucio.olivares@unicampania.it (L.O.); marco.vigliotti@unicampania.it (M.V.); daniela.ruberti@unicampania.it (D.R.)

[2] ISPRA—Institute for Environmental Protection and Research, 00144 Rome, Italy; paolomaria.guarino@isprambiente.it

[3] Independent Researcher, 81055 Santa Maria Capua Vetere, Italy; ing.robertopratelli@libero.it

[*] Correspondence: emilia.damiano@unicampania.it

**Abstract:** In northern Campania (Southern Italy), the historic center of many towns is characterized by the widespread presence of cavities in the subsoil, excavated over the centuries for quarrying tuff blocks for buildings, along with cathedrals, churches and chapels. A singular feature of these places of worship is, in fact, the presence of a wide and frequently connected network of underground cavities and tunnels, which were used for hydraulic, religious or connecting purposes. The cavity network is often unknown, abandoned or even buried, thus representing a risk for their susceptibility to sinkholes. Such elements are important as cultural heritage of inestimable value and as attractors for tourism; for this reason, the multidisciplinary study conducted on a place of worship in the Caserta area is illustrated herein: the Cloister of Sant'Agostino, in Caserta (XVI century CE). A geological and geotechnical characterization of the subsoil was performed at first. A laser scanner survey of the accessible cavities and the external churchyard was carried out. The resulting 3D model of the underground sector allowed for a clear understanding of the room size, their location, the levels and the path of the corridors. To understand the extension and layout of the crypts, Electrical Resistivity Tomography (ERT) surveys were undertaken in the surrounding areas. The analysis of the ERT measurements revealed some anomalies that could be ascribed to unknown structures (crypts). Finally, numerical methods were applied to estimate the stress state of the soft rocks and the potential areas of crisis, with preliminary assessments of the influence of the presence of cavities on the stability of the subsoil. The results allowed us to improve the knowledge of the study site and provide useful data for the planning of future targeted investigations, underlining how integrated research between applied disciplines can provide indispensable support both in the management and mitigation of geological risks in urban areas and in the sustainable reuse of hypogea.

**Keywords:** artificial cavities; Campania Grey Tuff; multidisciplinary characterization; Electrical Resistivity Tomography; stability assessment

## 1. Introduction

In recent years, multiple linkages between cultural heritage and geoheritage were identified and highlighted [1,2]. According to Ref. [3], culture and geological heritage are often integrated, especially in an urban context. Among the geoheritage issues in the latter context, the heritage stone represents a theme of interest with numerous examples from European cities [4], among others. The geological background of these cities can be read not only in famous historical buildings erected from characteristic stones but also from

the sites of stone extraction, often located within the cities or in the subsoil. In this sense, quarrying is part of cultural heritage, including excavation techniques, and the architecture of the hypogea, up to their subsequent reuse, is sometimes also converted into burial areas (under places of worship) and is therefore included in "cemeterial geotourism" [5,6].

From this perspective, in Italy, a strong effort was produced to provide a classification of the artificial cavities within the Italian Speleological Society [7]. The resulting Classification of Artificial Cavities was later adopted by the International Union of Speleology [8], thus attributing a cultural dimension to mining and quarrying that has received increasing recognition from the World Heritage Committee of UNESCO [2].

In northern Campania (Southern Italy), the historic center of many towns is characterized by the widespread presence of underground cavities. These environments testify to the multi-centennial exploitation of local geo-resources through the extraction of tuff rock, widely used in ancient, medieval, and modern-age constructions [9]. From ancient age structures to the seventeenth-century churches and palaces, passing through medieval convents and basilicas, Campania tuffs have been the primary material for the development of inhabited centers and related infrastructures. Among the latter, we can also include the same underground environments that responded to various social needs, such as the conservation of agricultural products, the burial of the dead, the cult of saints, and the general hydraulic needs of inhabited areas such as transport and storage of water, both rain and spring [10–17].

Through the years, the conversion of underground artificial cavities into tourist sites has increased, and these cavities, therefore, now recognized as an "underground heritage", represent a precious object of study and an irreplaceable testimony of the historical and deep-rooted relationship between humanity and their territory. Quarrying can indeed make connections with the local history by emphasizing the use of stone resources in buildings, including dwelling houses and conversion to other uses [18].

The protection and enhancement of the architectural, artistic and landscape heritage cannot therefore be separated from a precise and systematic knowledge of the underground environments of anthropic origin and, consequently, of the effects of the presence of these latter on the interconnected structures and buildings [19]. Although these empty spaces represent a cultural heritage of inestimable value, their distribution underground is still little known. Urban development has often sealed every signal of the presence of cavities, many of which were abandoned or even buried, to the point of representing a risk for their susceptibility to sinkholes [20–24]. This is even more dramatic when it happens in urban areas or underneath buildings of historical and cultural interest.

When compared to other geo-hazards, sinkholes are typically underestimated and difficult to predict. In this regard, an important issue is identifying possible precursory evidence of the phenomena and, therefore, the assessment of the degradation of rock and soil masses surrounding the caves and analysis of the weathering conditions, eventually leading, through a decrease in the physical and mechanical properties of the geomaterials, toward instability, collapse and sinkhole development [21,25,26]. The evaluation of the susceptibility to sinkholes is thus mandatory for the reuse of hypogea as tourist sites [27].

On this basis and considering the anthropic cavities an absolute documentary value, still unduly neglected, a multidisciplinary study was conducted on a place of worship in the historic center of Caserta (northern Campania), close to the well-known Royal Palace: the Cloister of Sant'Agostino (XVI century CE). This study was carried out by integrating research between applied disciplines (geology, geotechnics, speleology, cultural heritage) and providing indispensable support both for the management and mitigation of geological risks in urban areas and for the sustainable reuse of hypogea, thus contributing to enhancing the cultural and tourist promotion of a territory.

## 2. Study Area

The study area corresponds to the northern and northeastern part of the Campania Plain, in the northwestern part of the Campania Region (Italy; Figure 1). The plain is

considered a broad, complex graben controlled by NE–SW, NW–SE and E–W normal fault activity, established in the Late Pliocene [28] or the Early Pleistocene [29,30] along the Tyrrhenian side of the Apennine Mountains. The sedimentary evolution was mainly conditioned by the fluvial and marine processes and the volcanic activity of the Campi Flegrei, Somma-Vesuvius and Roccamonfina volcanoes [31–34].

The study area is characterized by a flat morphology, between 95 and 20 m a.s.l. The subsoil stratigraphic architecture is formed by a succession of different units composed of volcanoclastic deposits, in particular, related to the Campania Grey Tuff (CGT; 39 ky B.P; [35]) and Neapolitan Yellow Tuff (NYT; 15 ky B.P.; [36]) pyroclastic eruptions from the Campi Flegrei volcanic district [37].

The CGT deposits were spread over the whole Campania Plain, giving rise to a thick (up to 40 m thick), laterally continuous volcanoclastic unit characterized by different lithofacies, mostly derived from the different mineralogic compositions [38–40]. In this unit, in the Caserta area, Di Girolamo [38] distinguishes, from the base, three main facies (piperno, pipernoid tuff and gray tuff) with decreasing degrees of lithification and density; sometimes, completely incoherent facies ("cinerazzo") are present on top. The good mechanical characteristics of the tuff lithofacies justify the presence of numerous quarries and cavities, according to the availability of adequate thicknesses of coherent lithofacies [41,42].

This unit was covered by the surge deposits of the following eruption of the NYT and characterized, in the study area, by approximately 1 m of thick yellow-whitish and greyish ash in which centimeter-wide levels of whitish pumice are interspersed. Above this, a massive level of colluvium with whitish and yellowish cineric matrix is recognized, about 1 m thick, with scattered pumice, which gradually passes to the superficial pedogenized cover.

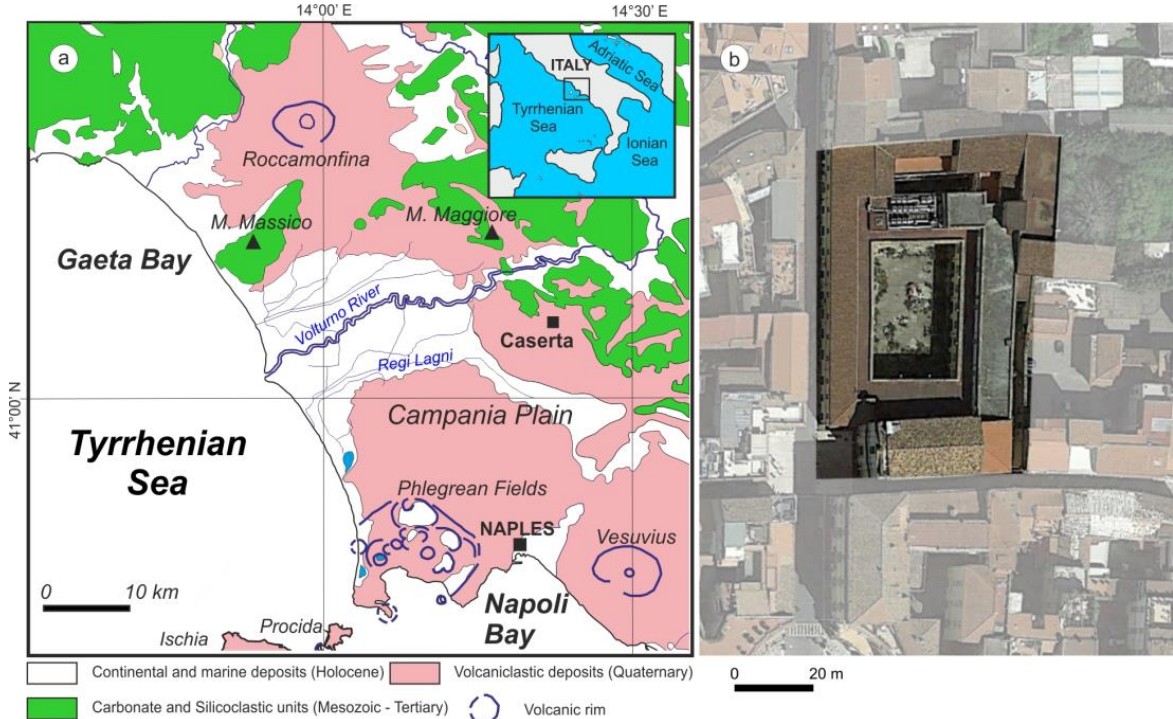

**Figure 1.** Location map of the study sites (**a**) Schematic geologic map of the study area (modified from [43]; (**b**) Cloister of Sant'Agostino in Caserta (14.331657° E, 41.072397°).

*The Cloister of Sant'Agostino*

The Sant'Agostino Complex is located in the historic center of the city of Caserta, very close to the garden of the Royal Palace (Figure 2).

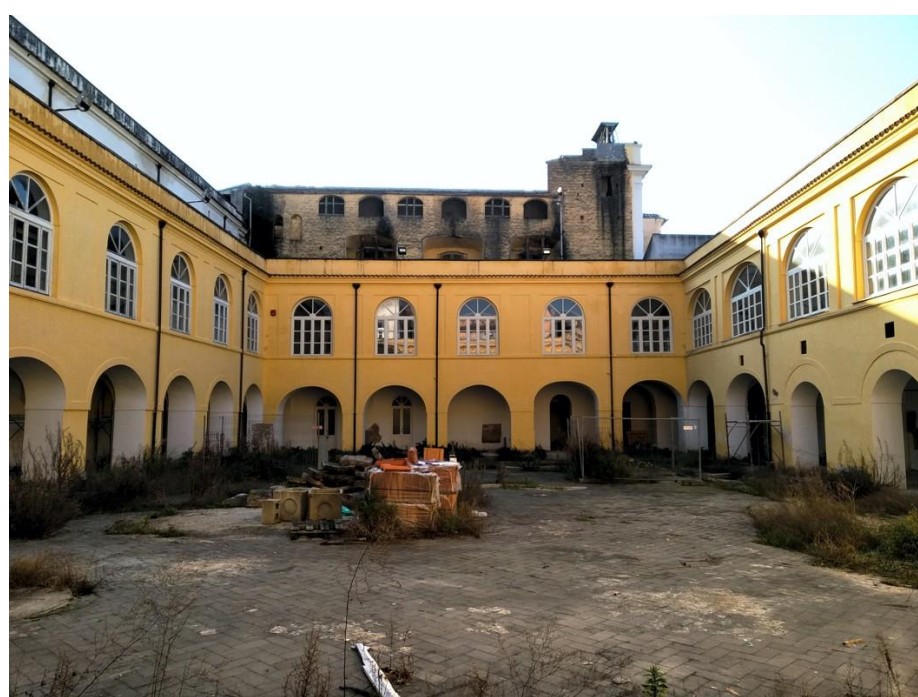

**Figure 2.** Detail of the courtyard of the Cloister of Sant'Agostino.

The historical nucleus around which the complex called "Convento (or Cenobio) of S. Agostino" develops dates back to the 13th century, and its current layout derives from a series of expansions and renovations that took place in the centuries following the first foundation, which are difficult to place with chronological precision. These modifications accompanied its transformation from an Augustinian convent, a function carried out until around the middle of the 17th century, to a boarding school for girls, an institution founded at the beginning of the 18th century.

The Church of Sant'Agostino houses seventeenth-century paintings, the Museum of Contemporary Art and the Museum of Traditions, both declared of regional interest.

The sixteenth-century internal cloister, with a rectangular shape, has dimensions of approximately 32 × 21 m. The development in elevation is on three functional levels, corresponding respectively to the ground floor, the first floor and the second floor, with the latter occupied partly by uncovered terraces.

Underlying the uncovered area of the cloister—and for approximately 2/5 of the corresponding surface area—there is an underground cavity of anthropic origin whose construction is probably to be considered contemporary with the first installation of the S. Agostino Complex.

The whole structure has recently been the subject of a restoration project that required investigations inside the cloister and, in particular, the cavity with an adjoining cistern. In the first instance, the local administration and the restoration project managers planned to fill the cavity with a mixture of cohesive soil. Then, the Superintendence of Cultural Heritage suggested preserving the hypogeum for its historical value and asked for a deep survey to assess the stability conditions.

## 3. Methods

### 3.1. Geological and Geotechnical Characterization

Geological and geotechnical characterization was carried out through comprehensive investigations to assess the state of stress induced by the presence of the cavities in the surrounding soils to preliminarily evaluate the stability conditions of the cavity's roof and individuate the associated potential risks. As a matter of fact, the stability of the cavity; the

compressibility of the soil covering the rock formation; and the presence, orientation and frequency of rock joints are all factors expected to influence the susceptibility to sinkholes.

The stratigraphic sequence and the mechanical characterization of the subsoil were derived by means of boreholes, penetrometer tests (DPSH types) and geophysical investigations (MASW type). First, the results of boreholes carried out in an area at most 300 m away from the investigated cavity, coupled with the results of indirect tests performed inside the cloister, were used to reconstruct the stratigraphic setting.

The state properties, the stiffness and the strength characteristics of the granular pyroclastic soils constituting the soft rock's cover were obtained by a joint interpretation of penetrometer and geophysical tests using the empirical correlations available in the literature for cohesionless soils [43,44] and the available data on pyroclastic soils of air-fall deposition widely diffused in the area [45–47]. Regarding the tuff formations, due to the lack of direct measurements, typical values reported in the literature for pyroclastic weak rocks [48–50] were adopted. At the present stage of the study, it was not possible to individuate the eventual presence of joints or discontinuities; therefore, the tuff rock formation was assumed to be continuous.

A numerical method was then applied to estimate the lithostatic stress field considering the presence of the empty spaces in the underground. The evaluation was performed by analyzing a 2D geometrical model based on the results of the topographical survey, using the finite element analysis. The model was limited in depth to 13 m below the floor of the cavity assuming a rigid base and considering exclusively the effect of the soil weight. Afterward, a preliminary assessment of the stability of the roof of the hypogeum was performed by using the stability chart proposed by Ref. [51] for anthropic cavities whose geometry resembles the investigated one.

### 3.2. Electrical Resistivity Tomography

In order to understand the extension and layout of the crypts, Electrical Resistivity Tomography (ERT) surveys were undertaken in the surrounding areas of the church.

Geoelectric surveys represent a modern methodology of non-invasive geophysical investigation and are based on the detection of electrical resistivity of the various types of land investigated [52]. This type of prospecting is very useful to identify the presence of cavities and to locate critical points for exploratory surveys, avoiding drilling in insignificant areas [53]. The test consists of placing on the ground a large number of electrodes connected to an instrument called a Georesistivimeter, able to acquire thousands of measurements that are properly processed by specific software (Res2DINV, Res3DINV ver.4.05.14) that provides a pseudosection, a qualitative representation of the electrostratigraphy of the subsoil. The result of a geoelectric survey is a 2D model of the subsurface, called electric tomography; it provides a prospect of the resistivity in the subsoil and allows to highlight local anomalies determined by resistivity values as ''too high" or ''too low" with respect to adjacent values.

In the case of the Sant'Agostino Cloister, six electric tomographies were made on the outdoor ground of the cloister and, specifically, 32 stainless steel electrodes were used and placed one meter from each one, for a total of 31 m. A dipole–dipole electrode configuration was used, sensitive to lateral variations, to investigate the electrical characteristics of the lithotypes present in the study area and to detect any anomalies of resistivity in the subsoil. The acquired data were processed by the RES2DINV software, which represents one of the most popular programs for the reversal of geoelectric data. The measure of the variations of the resistivity values enabled the definition of the geometry and the characteristics of the geological bodies and provided indications about voids in the subsurface.

### 3.3. Laser Scanner

A laser scanner survey of the cave networks extending beneath the Cloister of S. Agostino in Caserta was performed using a Leica BLK360 laser scanner.

Leica BLK360 works, as all laser scanner systems, based on the emission and subsequent receiving of laser beams sent towards the object to be detected, measuring the return time of the laser beam and, according to the time-of-flight TOF principle, measuring distance and angle of the reflection point.

The result of the scan consists of a point cloud, defined by their coordinates and by other parameters, i.e., the reflectivity coefficient k, which are useful for providing information aimed at discriminating different materials for porosity, compactness, etc. The instrument used is equipped with a photographic camera (the images are used to "texturize" the three-dimensional model obtained) and a thermal camera.

In this work, it was possible to carry out the survey without target positioning thanks to the redundant number of scans and the high precision of the instrumentation. Indeed, 12 setups in the case of the Cloister of S. Agostino were performed. Firstly, the results of the setups were used to reconstruct, through the software partner Register 360, a unique point cloud and the related 3D model of each cave network, allowing for the analysis of morphological and geological features. The performing of setups up to the surface allowed a detailed reconstruction, by means of orthographic and perpendicular projections, of relationships between underground cavities and buildings.

Lastly, the point clouds have been exported to other CAD software to enable (i) the comparison with the surveys performed with other methodologies; (ii) the processing of sections useful for stability analyses.

## 4. Results and Discussion

### 4.1. The Cloister of Sant'Agostino Architecture

The cavity below the cloister courtyard appears as a pentagonal excavation with a vault shape of a semicircular arch, which develops in a ring around a central cistern. Access is guaranteed by a staircase dug into the tuff, which is accessed from a room overlooking the portico (Figure 3). Three vertical access points (wells) are visible on the vault of the cavity, one on the stairs and, these days, sealed, and two at opposite corners of the cavity and visible in the courtyard. They represent the first access to the tuff, excavated from the ground surface into the upper loose pyroclastic deposits. Another vertical access is visible in the middle of the cistern vault and was likely used to obtain water.

The survey of the cavity with laser scanner techniques has highlighted the extension of a pentagonal-shaped cave, connected by means of vertical shafts to the courtyard above (Figure 4).

This survey provided rich geometric information in the form of a 3D cloud that offers an accurate three-dimensional restitution of the distribution of voids in the subsoil and their relationship with the buildings on the surface. In recent times, techniques based on mobile laser scanning systems have also found application in mining and quarrying [54], providing an interactive visual representation of varying geological conditions across the underground site. The point clouds thus obtained return a relief rendering reducing geometric uncertainties [55]. Furthermore, it must be considered that geometry is not the only feature required to produce hazard maps; the contribution of laser scanning also becomes significant in producing immediate 2D profiles that can be used for stability assessment discussed in the following paragraphs.

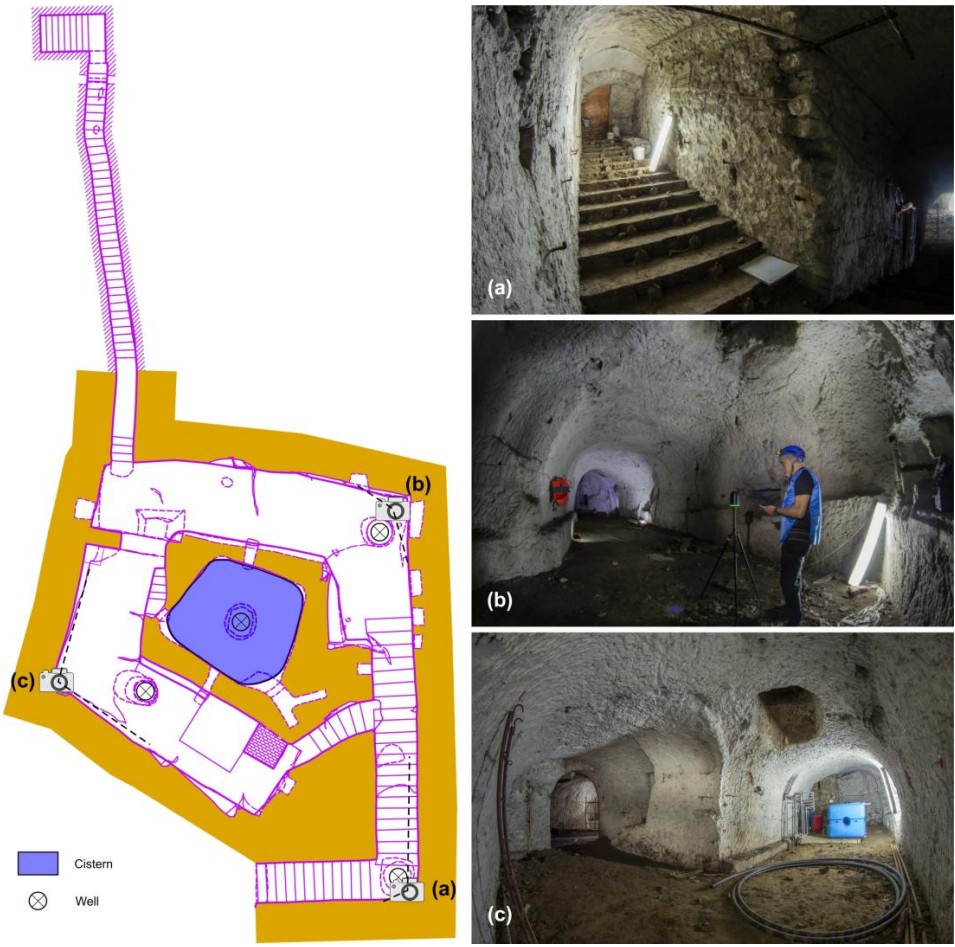

**Figure 3.** Plan view of the Cloister of Sant'Agostino and location of shooting points of parts of the hypogeum (letters refer to the photos on the right); (**a**) type of access: stair with three ramps; (**b**) view of the northeastern corner of the outer wall of the cistern; (**c**) view of the southwest corner with vertical access point.

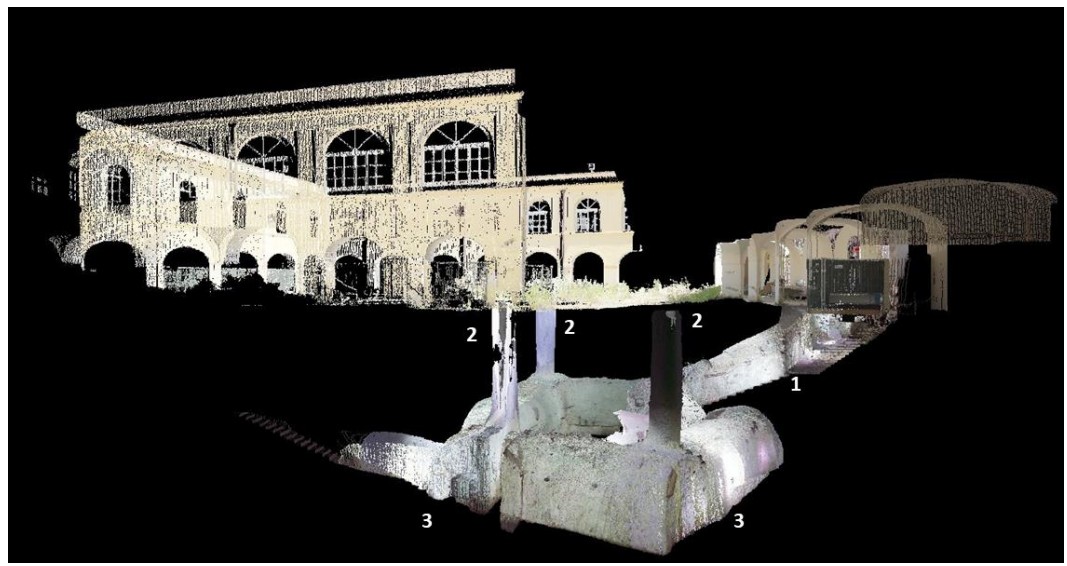

**Figure 4.** Sant'Agostino Cloister: relationship between the courtyard and the cave below. (1) Access; (2) shafts; (3) cave.

### 4.2. Geological and Geotechnical Characterization

The stratigraphic structure of the subsoil was reconstructed integrating bibliographic data deduced from geological investigations carried out in situ and in the surroundings. The first 4–5 m from the ground level are characterized by loose pyroclastic deposits, derived from eruptions after the emplacement of the CGT. The latter characterizes the underlying levels with the yellow lithofacies. The contact between the two units is marked by a thin layer of the so-called "cappellaccio", representing the upper portion of the lithoid tuff formation (CGT) altered in a subaerial environment. The transition to post-CGT deposits is marked by 1 m of palaeosoil, followed by loose ash, attributable to the NYT, intercalated with levels of whitish pumice. This is followed by colluvial deposits with a pyroclastic matrix, which, towards the top, become pedogenized. Pedogenic cover closes the succession.

This stratigraphic sequence is also confirmed by the analysis of the seismic and Dynamic Penetrometer Super Heavy (DPSH) tests performed inside the cloister. The low numbers of boulders measured during the penetrometer tests reveal the presence of very loose layers of granular soils, identified as Post-CGT in the first 4.5 m; at the base of this unit, a sudden increase in measurement parameters occurs, indicating a transition to the CGT formation (Figure 5).

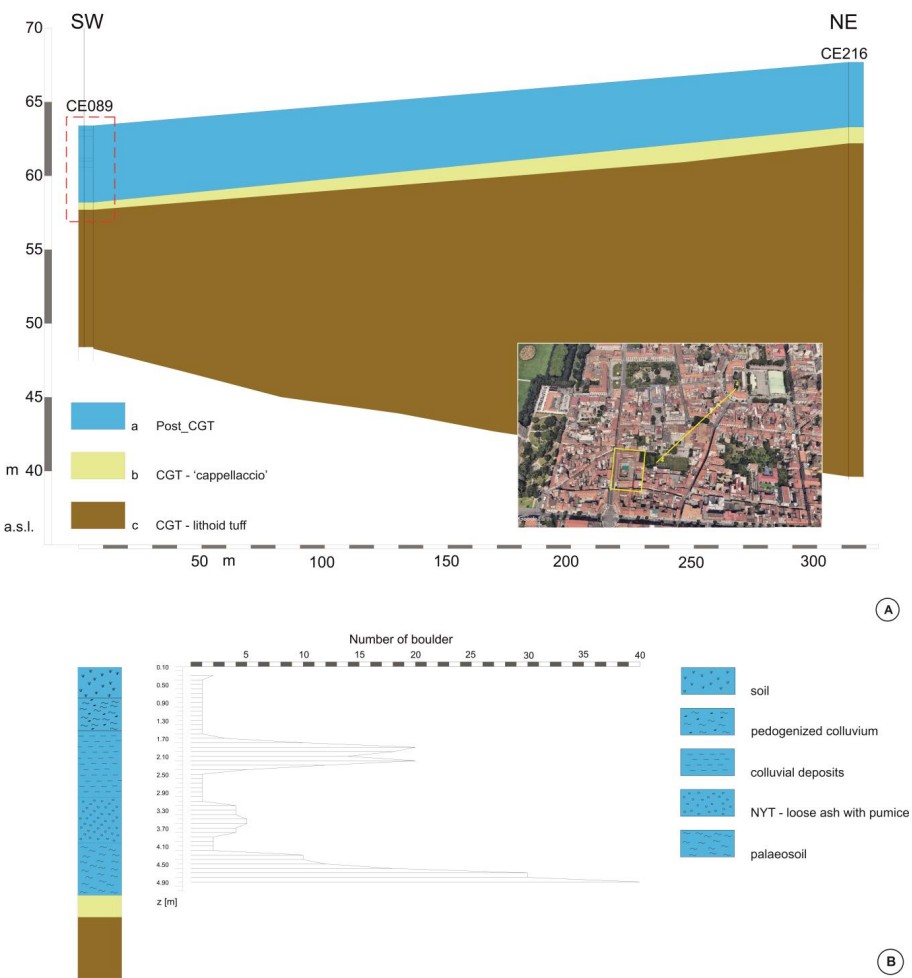

**Figure 5.** (**A**) Stratigraphic reconstruction of the geological setting of the study area, based on well log stratigraphies. In yellow, the track of the geological profile; the red dashed box refers to the detail in (**B**). (**B**) Detail of the stratigraphic setting of the Post-CGT deposits compared with the results of in situ DPSH investigations carried out in the Cloister of Sant'Agostino.

On the base of such results, the geotechnical model was derived, and the subsoil was schematized as a layer of cohesionless pyroclastic soil 4.5 m thick directly lying on the tuff formation (CGT). The parameters used for the rock and soil are shown in Table 1 in terms of unit volume weight ($\gamma$), Poisson coefficient ($\nu$), cohesion (c′), friction angle ($\varphi$′), uniaxial compression strength ($\sigma_c$) of the lithoid tuff and stiffness modulus (E). It has to be noted that, as the groundwater level in this area is very deep, the Post-CGT soils are usually unsaturated and the adopted value of cohesion (c′ = 15 kPa in Table 1) is not an effective cohesion, but it takes into account the beneficial effect of the unsaturated state on the shear strength of the soil [46,47]. Regarding the mechanical properties of the rock formation, both stiffness and strength parameters are the lower bounded values found in the literature [48–50] (Table 1).

**Table 1.** Cloister of Sant'Agostino: mechanical properties of the soils.

| Depth (m) | Soil | $\gamma$ (kN/m³) | $\nu$ | c′ (kPa) | $\varphi$′ (°) | $\sigma_c$ (MPa) | E (MPa) |
|---|---|---|---|---|---|---|---|
| 0–4.50 | Post-CGT | 15 | 0.30 | 15 | 35° | - | 170 |
| >4.50 | CGT | 14 | 0.30 | 270 | 35° | 1.07 | 620 |

### 4.3. Stability

Numerical analyses were carried out to preliminarily assess the lithostatic stress field in the subsoil, adopting a linear elastic constitutive model. The simplified 2D geometry illustrated in Figure 6 was considered, which represents the most critical situation due to the contextual presence of the central cistern and two lateral corridors (section A-A′ in Figure 3) in the hypothesis of continuity of the rock formation in the surroundings that, as shown in the following, needs to be verified. Initial conditions were derived under the assumption of a lithostatic state in the absence of the cavity, assuming a value of the coefficient of earth pressure, $k_0$, equal to $\nu/(1 - \nu)$ for both the cohesionless soils and the tuff. Subsequently, the formation of the cavity was simulated by the excavation of the vertical wells starting from top to bottom, as normally performed during tuff's mining processes.

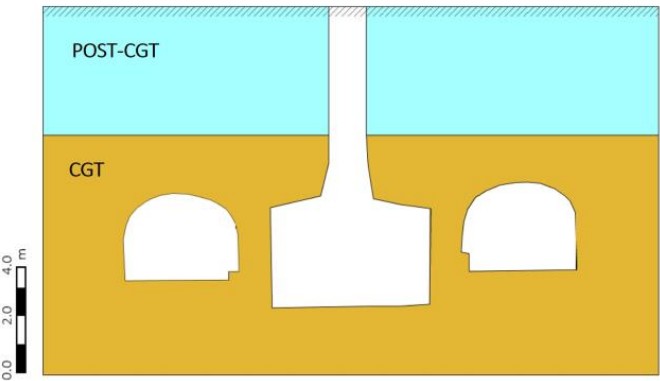

**Figure 6.** Modeled section A-A′.

The lining of the walls of the vertical extraction shafts was not considered in the present analysis, as this is a recurring condition for the hypogeum in the Campania plain, as well as for the studied one. The presence of unlined shafts represents a potential risk for the fruition of the cloister and the underground cavities system. Moreover, in the case of the Sant'Agostino Cloister, some of the unlined wells are also exposed to rainwater infiltration directly coming from the disconnected cloister pavement. Consequently, the unsaturated pyroclastic soils crossed by the vertical shafts can undergo saturation, which, in such loose materials, is accompanied by volumetric collapse and vanishing of the apparent cohesion up to soil erosion and caves, thus increasing the risk of instability of the subsoil.

At the end of the excavation phase, as expected, the state of stress in the subsoil was deeply altered (Figure 7). The tensional release induced by the presence of the empty spaces

gives rise to shear stress values as high as 380 kPa even due to the implemented simplified sharp-edged geometry (the cistern is not accessible, and its geometric survey has been carried out from two windows located on its lateral walls).

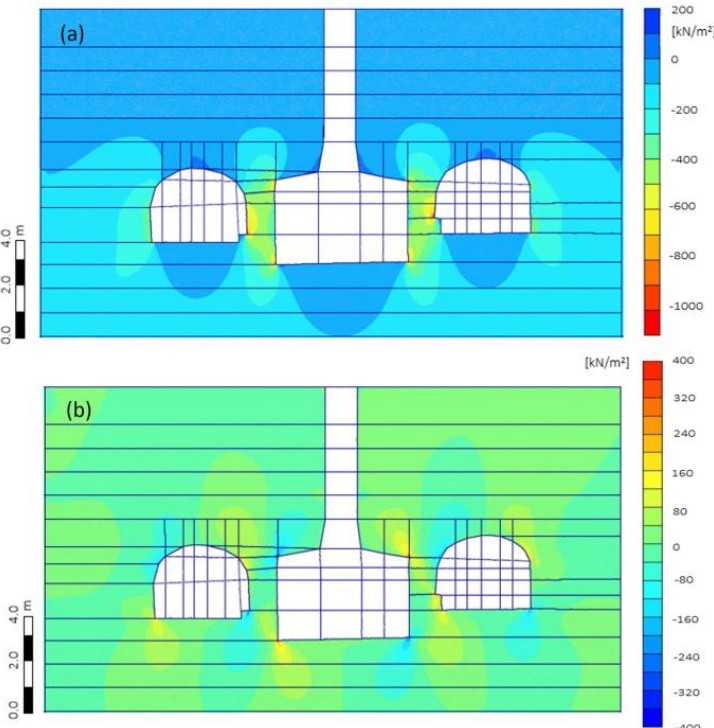

**Figure 7.** Stress field at the end of excavation phase: (**a**) vertical effective and (**b**) shear stresses.

The results of the numerical analysis were used to preliminary assess the stability condition of the cavity's roof. As, at the present stage of this study, experimental data on strength characteristics of the CGT formation and their spatial availability are not available, the evaluation was performed by using the stability chart proposed by Ref. [51]. The curves reported in the chart and illustrated in Figure 8 express the ratio between the minimum rock uniaxial compression strength ($\sigma_{c,min}$) ensuring the static equilibrium and the acting vertical stress ($\sigma_v$) as a function of the cavity geometric characteristics (ratio between cavity span, L, and rock cover thickness, t), and they were determined by means of 2D numerical analysis. In the frame of the reliability-based assessment, curves associated with three different probabilities of failure (1%; 10%; 50%), related to the possible random variability of rock strength and horizontal joint location in the roof, were calculated by the authors, which bound four different stability zones. In the chart (Figure 8), the conditions of the cistern and lateral corridor roofs in the Sant'Agostino hypogeum, estimated by considering the ratio between the rock uniaxial compression strength ($\sigma_c$) (Table 1), assumed homogeneous, and the acting vertical stress ($\sigma_v$) are represented by the black dots. All the dots fall within the "attention" and "critical" zones, indicating that, although the estimated safety factor (minimum vertical distance between the dots and the black line in Figure 8) is around 1.5, the uncertainties about the mechanical properties of the rock mass and the adopted simplified 2D analysis, which strongly affect the evaluation, need to be reduced, and more refined tools are needed to evaluate the actual stability condition of the cave. This can be performed only after acquiring clear and exhaustive knowledge of the extension of the hypogea and adequate mechanical characterization of the soft rock in which it was dug.

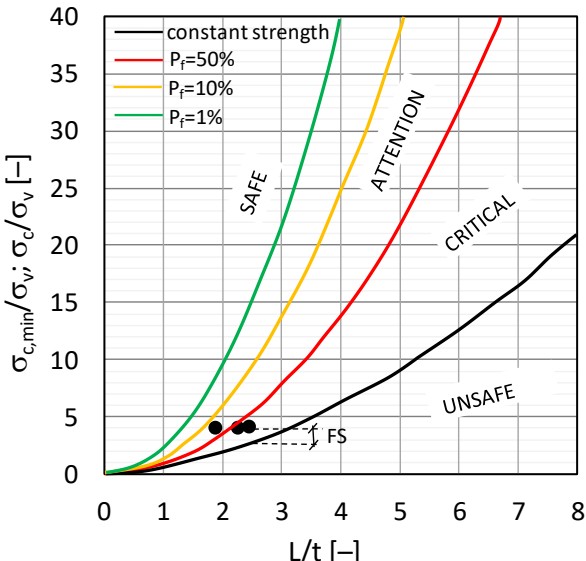

**Figure 8.** Simplified assessment of the cistern's roof stability based on the stability charts proposed by Ref. [51].

### 4.4. ERT

The use of geophysical investigations is considered the best method of finding and outlining underground cavities [56,57]. Among them, ERT investigations provide a low-cost, fast and robust investigation tool.

Given the complexity of the monumental complex, the presence of other voids in the area was suspected. The electrical tomography investigations for the Sant'Agostino Cloister were performed to understand the real extension of the cavity body and to confirm or deny the presence of other voids.

The six arrays of electrical tomography, positioned on the pavement of the courtyard of the cloister, resulted in six sections showing the 2D distribution of the electrical resistivity of the investigated terrain (Figure 9). By associating the lithology with the resistivity values obtained, it was possible to obtain the following general table:

- Blue electrical resistivity between 3.19 and 7.94 ohm.m, associated with the presence of infiltration water;
- Light blue electrical resistivity between 7.94 and 19.8 ohm.m, associated with humidified pyroclastics, often humified (pedogenized), with slightly compacted silty grain size distribution;
- Electrical resistivity between 19.8 and 123 ohm.m of light green and dark green colors, associated with medium compacted reworked sandy pyroclastics;
- Electrical resistivity between 123 and 766 ohm.m of yellow, ocher-yellow, orange, red and dark red color, associated with dense sandy pyroclastics; they correspond to reworked materials in the more superficial layers;
- Bordeaux red electrical resistivities higher than 766 ohm.m could be associated with very compact gray tuff or voids due to cavities.

On the whole, in all the performed tomographies, a first layer is found in the investigated subsoil starting from the ground level, characterized by low values of electrical resistivity, which can be associated with sandy pyroclastites arranged in a chaotic and non-stratified way, and a second layer characterized by higher values of electrical resistivity associated with the presence of tuff and underground voids. These latter are evident in sections DD' and CC' and, in this case, should correspond to the detected cavity, the access shaft and the presence of the stairwell to the north.

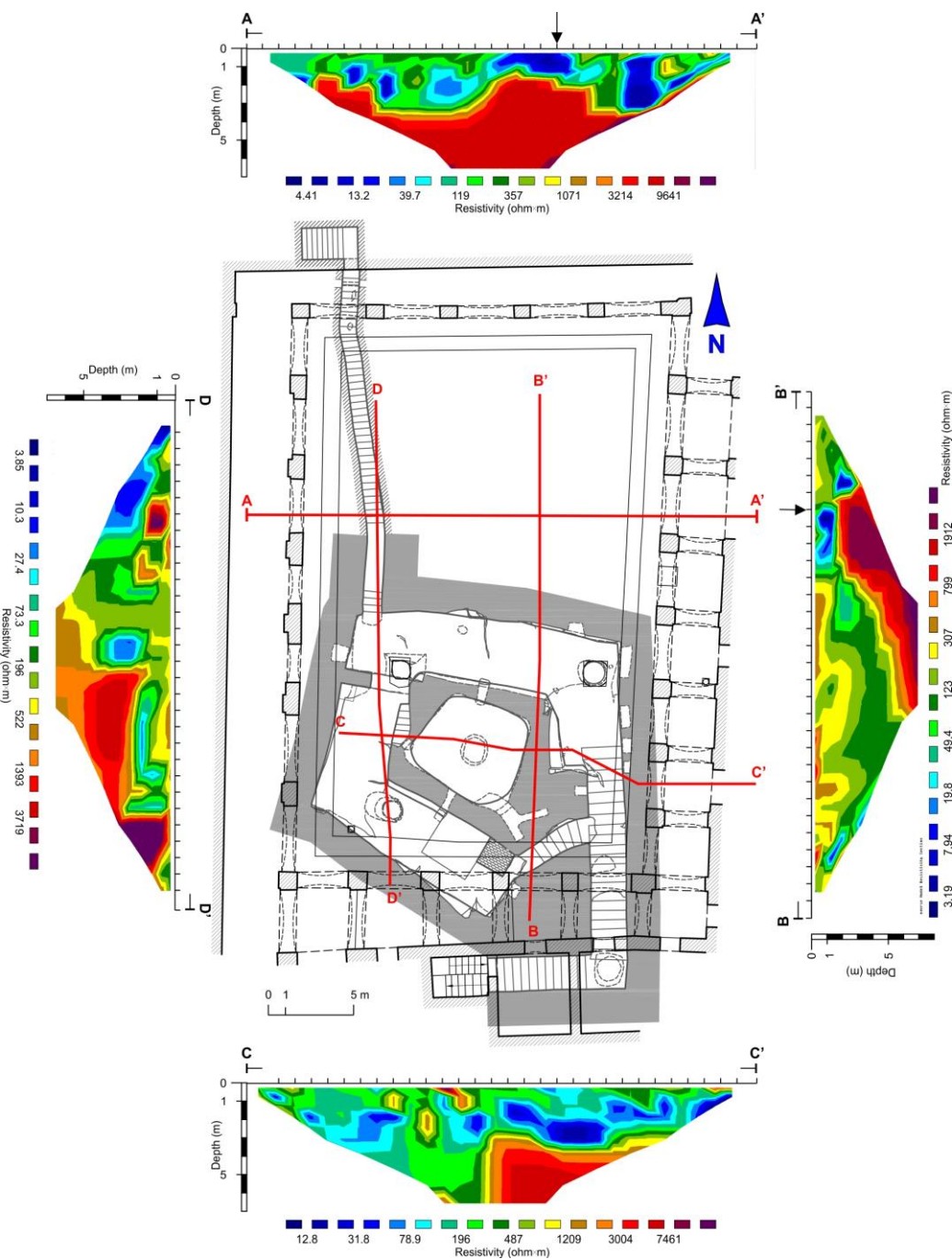

**Figure 9.** Plan view of Cloister of St. Agostino and the map of the cavity, the cistern and the vertical shafts below the courtyard. Red lines indicate the ERT profiles traces. On top and left of the resistivity profiles, the black arrows indicate the intersection of the two profiles. For explanation, refer to text.

The tomography thus confirmed the presence of the known and accessible cavity. In the initial hypothesis, the presence of underground voids was also suspected in the northern part of the cloister courtyard, and the results obtained would seem to confirm the existence of an inaccessible empty space. In fact, in section BB' (Figure 9), there are strong contrasts of electrical resistivity starting from a depth of about 2 m, rather superficially, considering that in the study area, the tuff notoriously characterized by high resistivity values is present at a depth of approximately 4 m, and it is excluded that these resistivities can be attributed to the tuffaceous formation. All this suggests a possible existence of an underground void, also confirmed by the profile AA', which intersects the previous

one, reporting anomalous resistivity values. In fact, the north stairwell has a part of a walled-up side wall in its intact part, as if to signify a sealing of a possible connection to the inaccessible void. However, the accessible cavity does not allow for reaching and inspecting this part of the subsoil to be identified, though it would be advisable to carry out a coring.

### 4.5. Discussion

The results described above provide knowledge of an underground space in an urban area under a worship place of high historical interest. This study aimed to address the restoration and conservation of the underground heritage by characterizing the underground space with a multidisciplinary approach. This will help to disseminate knowledge on underground culture from many different aspects (i.e., history, geotechnics, geology and architecture, among others) and contribute to the EU COST Action Underground Built Heritage as a catalyzer for Community Valorization (Underground4value) that aims to promote the preservation and valorization of Underground Built Heritage [58]. Underground4value has made significant contributions to balanced and sustainable approaches for this purpose with over 200 experts from several countries collaborating, sharing methodologies, case studies and best practices with an interdisciplinary focus. There are several challenges to making use of underground spaces, and one of the most important is that we often lack sufficient knowledge of these sites.

From this point of view, the individual techniques used in this study to characterize the cavity present under the Cloister of S. Agostino are currently used in this type of study, but in a few cases, however, we observe their use in such a highly integrated manner. The application of numerical modeling to the stability analysis of underground cavities is increasing due to the availability of numerical models [59,60] that overcome those that assume simplified geometrical, geological and geomechanical conditions. In this perspective, the detailed geometrical feature reconstruction provided by the laser scanning survey provided a 2D and 3D geometric characterization that first of all allowed us to identify the exact position of the voids with respect to the surface and foundations of the cloister; this method is not affected by the geology around the cavity and obtains a very clear visual 3D model of the cavity in a very short time. Furthermore, the point cloud produces immediate 2D profiles to be used for stability assessment [54]. The integration with the ERT technique adds further potential in the characterization of cavities: if on the one hand, it has proven to be a versatile and reliable tool for mapping subsurface features like voids, cavities or tunnels, on the other hand, it has allowed us to broaden the knowledge of the underground space in a larger area than the one investigated, highlighting other voids not yet known. Furthermore, the approach used in this study yields fruitful information for further historical and/or archaeological survey design and for the interpretation of ERT investigations targeting similar geological features and structures [61].

### 5. Conclusions

This present study showed that the knowledge of a cavity system and the identification of the potential risks associated with its abandoned or unknown state is a fundamental step in the management of the geological hazards in urban areas and can be significantly improved by using a multidisciplinary approach that combines experimental and numerical investigations. This allows us to deepen the knowledge about the geometrical and mechanical features of the cavity and the above soils, to preliminary assess their stability and to detect the potential presence of unknown subsoil spaces close to the accessible ones. In this regard, of main relevance was the assessment of the reliability and potentiality of the geophysical surveys that, in addition to providing an overall characterization of the site of interest, can detail the presence of a network of cavities, and guide other types of investigations that integrate the achieved results. Among these, as also indicated by the geological–geotechnical investigations, there is the need to extensively characterize the tuff formation from a mechanical point of view and to refine the simplified 2D numerical

analysis, taking into account the real geometry of the empty spaces, to be able to evaluate the actual stability condition of the cave, especially when the reuse of the hypogea and the opening of the cloister to the public is planned. All the gathered information provides useful indications for the planning of future targeted investigations.

Above all, this study underlines how integrated research between applied disciplines (in this case, geology, geotechnics and topography) can provide indispensable support both for the management and mitigation of geological risks in urban areas and for the sustainable reuse of hypogea, thus contributing to enhancing the cultural and touristic promotion of a territory.

**Author Contributions:** Conceptualization and methodology, E.D., D.R. and L.O.; validation and formal analysis, all the authors; investigation, M.A.F., M.V., R.P. and P.M.G.; data curation, M.A.F. and E.M.; writing—original draft preparation, E.D. and D.R.; writing—review and editing, all the authors; visualization and supervision, all the authors; funding acquisition, D.R. and L.O. All authors have read and agreed to the published version of the manuscript.

**Funding:** The research activities were partially funded for D.R. by the Collaboration Research Program on "Census, analysis and evaluation of the Cavity System" present in the territory of the Hydrographic District of the Southern Apennines (Italy). This research was also funded by the V:ALERE 2020 Program (VAnviteLli pEr la RicErca) of the University of Campania "Luigi Vanvitelli", DDG n. 516–24/05/2018 and by an Accord with the Municipality of Caserta, Italy.

**Data Availability Statement:** The data used in this research work are available upon request from the corresponding authors.

**Acknowledgments:** The authors would acknowledge the municipality of Caserta for access to the cloister and the cavity.

**Conflicts of Interest:** The authors declare no conflicts of interest.

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
