# Peer review of "A Multidisciplinary Approach for the Characterization of Artificial Cavities of Historical and Cultural Interest: The Case Study of the Cloister of Sant’Agostino—Caserta, Italy"

_geosciences, doi:10.3390/geosciences14020042_

Round 1

Reviewer 1 Report

Comments and Suggestions for Authors

Dear Authors,

It has been a pleasure for me to review your manuscript “A multidisciplinary approach for the characterization of artificial cavities of historical and cultural interest: the case study of the Cloister of Sant’Agostino - Caserta, Italy”.

This is an interesting manuscript with well-organized data, gets straight to the point and is easy to understand for all types of readers.

The aim of this study is an integrate research between applied disciplines (geology, geotechnics, speleology, cultural heritage) which provide indispensable support both for the management and mitigation of geological risks in urban areas and for the sustainable reuse of hypogea thus contributing to enhancing the cultural and tourist promotion of a territory.

More in detail

The "Abstract" properly resumes the study approach and the obtained results.

The "Introduction" paragraph is concise and goes straight to the focus of the manuscript. In my opinion, the authors should cite some recent works, such for example:

- Varriale, R., Parise, M., Genovese, L., Leo, M., Valese, S. 2022. Underground Built Heritage in Naples: From Knowledge to Monitoring and Enhancement. In: D'Amico, S., Venuti, V. (eds) Handbook of Cultural Heritage Analysis. Springer, Cham. https://doi.org/10.1007/978-3-030-60016-7_70.

- Varriale R. “Underground Built Heritage”: A Theoretical Approach for the Definition of an International Class. Heritage. 2021; 4(3):1092-1118. https://doi.org/10.3390/heritage4030061

In the paragraph “Study area” authors briefly describe the geomorphology of this area; they should spend some lines describing the cited lithologies more in detail. They also should add some lines to improve a bit the description of the Cloister of Sant'Agostino.

The paragraph “Methods” clearly describes the examination methods adopted in this study.

The results are presented rationally, and the data, which seem to have been produced very carefully, are well presented and discussed. The title of the sub-paragraph 4.2 is “Geological and geotechnical characterization” but the authors did not present any geological or petrographical data of the rocks. They cite some “…bibliographic data deduced from geological investigations carried out in situ…”, I would be grateful to the authors if they could add at least a sketch map or a schematic map of the rock formations in this area.

I also suggest correcting the title of paragraph 3 in “Result and Discussion”, considering that the authors did not present only the results, but they discussed them.

The conclusions of this paper are well supported by appropriate evidence and provide good answers to the aims of the study.

The references cited in this paper are recent and relevant, correctly organized, and support the statements in the manuscript. Pictures and diagrams are of good quality and are readable.

In conclusion, this manuscript highlights the importance of research questions and outlines the study's approach. Combined experimental and numerical investigations at the site significantly improved our understanding of the cavity, and surrounding soils, and identified potential risks in its abandoned state. Particularly, geophysical investigations validated their reliability and detailed underground spaces, guiding further inquiries. Emphasizing interdisciplinary collaboration in geology, geotechnics, and topography, this study offers crucial support for managing geological risks in urban areas. Furthermore, it showcases the potential for sustainable repurposing of underground spaces, contributing to cultural enrichment and tourism promotion in a territory.

Reviewer 2 Report

Comments and Suggestions for Authors

At page 7, in figure 5, could be useful to insert a legend for tre three lithology colours a, b,and c

Author Response

Reviewer comment n. (1)        At page 7, in figure 5, could be useful to insert a legend for the three lithology colours a, b,and c.

Response: We acknowledge the reviewer for the suggestion. Figure 5 has been modified accordingly.

Reviewer 3 Report

Comments and Suggestions for Authors

The paper investigates with an interesting topic. However, I think that it is very local and maybe the overall interest is restricted. The presentation of the analysis is good but I would like to point out important deficiencies in terms of the deriving results. Although you mention that this research provides tools for future relative planning you don't explain these tools. How this spesific reserach has led to spesific tools? How these tools will be implemented to future investigations? How the results bring an overall feedback in relation, probably, with a network of caves in the area? I suggest that this topics need to be analysed in the conclusions and also in the introduction part, in relation to other established literature on these as well.
